# Laboratory studies on the optical, physical, and chemical properties of fresh and aged biomass burning aerosols

Zheng Yang[1], Qiaoqiao Wang[1], Qiyuan Wang[2], Nan Ma[1], Jie Tian[2], Yaqing Zhou[1], Ge Xu[3], Miao Gao[3], Xiaoxian Zhou[1], Yang Zhang[2], Weikang Ran[2], Ning Yang[1], Jiangchuan Tao[1], Juan Hong[1], Yunfei Wu[4], Junji Cao[4], Hang Su[4], Yafang Cheng[5]

[1]Guangdong-Hongkong-Macau Joint Laboratory of Collaborative Innovation for Environmental Quality, Institute for Environmental and Climate Research, College of Environment and Climate, Jinan University, Guangzhou 511443, China

[2]State Key Laboratory of Loess and Quaternary Geology, Institute of Earth Environment, Chinese Academy of Sciences, Xi'an 710061, China

[3]Xi'an Institute for Innovative Earth Environment Research, Xi'an 710061, China

[4]Institute of Atmospheric Physics, Chinses Academy of Sciences, Beijing 100029, China

[5]Minerva Independent Research Group, Max Planck Institute for Chemistry, Mainz 55128, Germany

*Correspondence to:* Qiaoqiao Wang (qwang@jnu.edu.cn) and Qiyuan Wang (wangqy@ieecas.cn)

**Abstract.** Atmospheric brown carbon (BrC) plays a significant role in global warming, yet the evolution of its optical properties during aging remains poorly understood, leading to substantial uncertainties in its climate effects. In this study, we investigate the aging process of BrC and its driving factors using laboratory-generated biomass burning emissions, including four types of straw and one type of wood. Upon OH oxidation, there exists a large increase in OA fraction after 2-day aging, followed by a minor increase during aging to 7 days. The particle growth is dominated by the change in OA content and thus shows a similar trend during aging. The mass absorption efficiency (MAE) of fresh BrC measured at 370 nm is 2.1–5.7 $m^2 \, g^{-1}$. A sharp decline in MAE is observed after 2-day aging, equally attributed to photobleaching and secondary organic aerosol formation. Although a negative correlation is observed between particle size and MAE, the reduction in MAE is mainly driven by the decline in the imaginary part ($k$) of BrC, with particle size playing a minor role. Combined with positive matrix factorization (PMF) analysis, the study reveals that oxygenated OA, characterized by higher O/C ratios but lower MAE, increases significantly with aging. In contrast, two hydrocarbon-like OA factors with lower O/C ratios and higher MAE, decrease over time. These results emphasize the importance of categorizing BrC based on its MAE and atmospheric behavior in climate models.

# 1 Introduction

As a special class of organic aerosol (OA), brown carbon (BrC) exhibits a significant light absorption ability at near-UV and shorter visible wavelengths with a strong wavelength dependence (Laskin et al., 2015). It accounts for approximately 19–40% of the total absorption by carbonaceous particles (Feng et al., 2013; Lin et al., 2014), with a global radiative forcing ranging from +0.03 to +0.57 W m$^{-2}$ (Li et al., 2023a). BrC can also perturb the temperature structure of the atmosphere and then influence cloud cover, known as semi-direct effects (Laskin et al., 2015; Yan et al., 2018). When deposited on snow and ice, BrC can reduce surface albedo and cause early snow melting (Qian et al., 2015; Tuccella et al., 2021). Additionally, the strong UV-absorption due to BrC can reduce the rate of atmospheric photochemical reactions, resulting in a 15%–30% decrease in regional concentrations of OH radical and O$_3$ (Wang et al., 2022). However, current understanding of the BrC light absorptivity, especially its evolution upon aging processes, remains limited, driving a significant uncertainty in the estimates of its climate effects.

BrC originates from a variety of sources, including primary emissions from biomass burning, coal combustion and vehicular emissions (Olson et al., 2015; Sun et al., 2017; Xie et al., 2017a; Park et al., 2018), as well as secondary processes such as photochemical oxidation (Lambe et al., 2013; Xie et al., 2017b), aqueous-phase processes (Lin et al., 2015; Ye et al., 2019) and nighttime oxidation (Jiang et al., 2019; Chen et al., 2023). Among these sources, biomass burning has long been recognized as a dominant source (Saleh, 2020). When considering BrC light absorption, the overall regional or global radiative forcing of biomass burning aerosol can shift from negative to positive effects (Saleh et al., 2015).

Many laboratory and field studies have investigated the chemical and optical properties of BrC from biomass burning (Hems et al., 2021). The optical properties of freshly emitted BrC, such as the absorption Ångström exponent (AAE) and mass absorption efficiency (MAE) or imaginary part ($k$), have been shown to strongly depend on burning conditions, with BrC from flaming biomass burning combustion exhibiting higher MAE and lower AAE than that from lower-temperature smoldering combustion (Laskin et al., 2015; Saleh, 2020). Utilizing the ratio of BC/OA as a proxy of the burning condition, several studies have established a quantitative relationship between the MAE and the burning conditions based on laboratory combustion experiments (Saleh et al., 2014; Xie et al., 2017a; Park et al., 2020). However, the relationship varies with different studies, and is suggested to be affected by the biomass types (Xie et al., 2017a; Park et al., 2020). So far, to what extent could the

biomass types affect the BrC absorptivity is not clear and requires further study.

Upon aging processes, BrC absorptivity could decrease significantly due to the photobleaching of some chromophores, with a lifetime ranging from a few hours (Zhao et al., 2015; Browne et al., 2019) to a few days (Forrister et al., 2015; Sumlin et al., 2017). The photobleaching rate largely depends on ambient conditions, including concentrations of OH radical (Wang et al., 2014) and $NO_x$ (Yang et al., 2022) as well as relative humidity and temperature (Klodt et al., 2023; Gao et al., 2024). The chemical characteristics of BrC also determine the extent of the photobleaching effect. For example, tar balls from biomass burning are found to be resistant to the photobleaching process (Saleh, 2020). Moreover, the light absorption of BrC can be enhanced through the functionalization and polymerization of existing OA (Wong et al., 2019; Hems et al., 2020) or the formation of new nitrogen-containing organic compounds, e.g., nitroaromatics (He et al., 2022; Yang et al., 2022). The photobleaching and enhancement may occur concurrently, making the evolution of BrC absorptivity more complicated. Some studies reported continuously decrease in $k$ (Liu et al., 2021) while some show a slight increase first followed by a significant decrease (Cappa et al., 2020; Schnitzler et al., 2020). However, current studies mainly focus on the evolution of the overall BrC absorptivity and few have endeavored to distinguish the behaviors of different BrC components (Wong et al., 2019; Fleming et al., 2020), which may undergo totally different aging processes.

In this study, we characterized the optical, physical, and chemical properties of fresh and aged BrC emitted from biomass combustion. By analyzing the synchronous evolution of both chemical components and light absorption in the smoke, we explored the aging processes of different BrC components and their contributions to the overall light absorption at different aging levels. The study demonstrated significant discrepancies in the aging processes among different BrC components and suggested the necessity to classify BrC based on its optical properties, especially its photobleaching rate, for its better representation in climate models.

## 2 Materials and methods

### 2.1 Experimental setup

Five types of biomass fuels were collected from major crop producing areas in China, including wheat straw (WS), rice straw

(RS), corn straw (CS), soybean straw (SS) and apple branch (AB) (Table S1 and Fig. S1). The rice is mainly distributed in central and southern China, and others are mainly distributed in northern China. These biomass types could represent the majority of China's biofuels. The combustion experiments were conducted at the Institute of Earth Environment, Chinese Academy of Sciences (IEECAS) in Xi'an, China. Figure S2 illustrates the instrument configuration. Emissions were generated by burning batches of ~10 g of dry biomass fuels (cut into pieces of 10–15 cm) on a combustion platform in a ~8 $m^3$ combustion

chamber. After the flame was extinguished, the smoke was first mixed and allowed to stand in the combustion chamber, and was then diluted before being sampled by several online instruments. Details of the chamber and combustion facilities are provided in Tian et al. (2015).

A Potential Aerosol Mass-Oxidation Flow Reactor (PAM-OFR) (Aerodyne Research, LLC, Billerica, MA, USA) was applied to simulate atmospheric aging processes. For each test, the fresh (F) aerosols were measured in the first 40 minutes and then the UV light of PAM-OFR was turned on for the next 40 minutes to measure the aged aerosols. Detailed information

of the PAM-OFR is described by Cao et al. (2020). Briefly, inside the PAM-OFR, OH radicals were formed through a series of photochemical reactions of $H_2O$ and $O_2$ under 185 nm UV illumination. The simulated OH concentrations can thus be controlled via adjusting the UV light intensity. In this study, two aging levels were simulated by applying two distinct UV intensities. For each aging level, the UV intensity was kept constant throughout the experiment by regulating the lamp voltage.

Based on the residence time within the PAM-OFR (90 s) and an assumed average atmospheric OH concentration of $1.5 \times 10^6$ molecules $cm^{-3}$ (Mao et al., 2009), the equivalent atmospheric aging levels were estimated to be around 2-days (A-2) and 7-days (A-7) in this study, similar to those reported by Li et al. (2020) and Guo et al. (2022). Table S1 also summarizes the conditions for all burning tests in this study. It is important to note that the experimental conditions do not perfectly represent the photochemical conditions of the atmosphere, and it emphasizes OH-driven oxidation under initially high-$NO_x$ conditions

which rapidly shift towards low-$NO_x$ conditions (Cappa et al., 2020).

**2.2 Gas analysis**

The CO and $CO_2$ concentrations were monitored using a Fourier Transform Infrared (FTIR) gas analyzer (DX4015, Gasmet, Finland). Before each group of experiments, we cleaned the FTIR sample cell with nitrogen and performed background

measurements. Gas concentrations were treated by FTIR standard procedures (Calcmet v12.15) using the linear relationship between absorbance and molecular number combined with the reference spectra. The modified combustion efficiency (MCE), defined as $\triangle CO_2/(\triangle CO_2+\triangle CO)$, was used to indicate the burning conditions during each fire test (Akagi et al., 2011; Wang et al., 2020b; Zhao et al., 2022). Here, the $\triangle CO$ and $\triangle CO_2$ represent the background-corrected CO and $CO_2$ values in the smoke. The MCE measured in this work ranged from 0.95 to 0.99 (Table S1). An MCE value greater than 0.9 is indicative of flaming combustion (Sinha et al., 2003; Akagi et al., 2011). The $NO_x$ and $SO_2$ concentrations were monitored both before and after the PAM-OFR (Fig. S2) using $NO_x$ analyzers (model 42i, Thermo Scientific Inc., USA) and $SO_2$ analyzers (model 43i, Thermo Scientific Inc., USA), respectively. The consumption rate of $NO_x$ and $SO_2$ during the aging process could then be derived base on their concentrations measured after the PAM-OFR relative those before the PAM-OFR.

**2.3 Aerosol characterization**

The mass concentrations of non-refractory chemical components were measured by a Time-of-Flight Aerosol Chemical Speciation Monitor (ToF-ACSM; Aerodyne Research Inc., USA), including organics, nitrate ($NO_3^-$), sulfate ($SO_4^{2-}$), chloride ($Cl^-$) and ammonium ($NH_4^+$). A detailed description of this instrument can be found in Fröhlich et al. (2013) and Xu et al. (2017). Briefly, the aerosols were first dried with a diffusion dryer and then passed through a critical orifice into a narrow beam via an aerodynamic lens. The aerosols were successively vaporized by a heated surface (~600 ℃), ionized via electron ionization, and detected by a mass spectrometer detector. The collection efficiency (CE) value was 0.5 in this study (Middlebrook et al., 2012). The calibrations were performed by dried monodispersed (300 nm) ammonium nitrate and ammonium sulfate particles. The relative ionization efficiencies (RIEs) of 3.93 and 0.82 were used for ammonium and sulfate and the default values of 1.1, 1.4 and 1.3 were used for nitrate, organics and chloride, respectively (Jimenez et al., 2003; Canagaratna et al., 2007). The ToF-ACSM data were analyzed with the standard data analysis software (Tofware v3.3) within Igor Pro (v7.08; WaveMetrics, Inc., Oregon, USA). In addition, Positive matrix factorization (PMF) (Paatero and Tapper, 1994) was performed on the high-resolution mass spectral matrix of OA (Ulbrich et al., 2009; Zhang et al., 2011). Finally, three OA factors were identified by the PMF model, including two hydrocarbon-like OA (HOA-1 and HOA-2) and one oxygenated OA (OOA).

To quantify the photochemical effect on aerosol chemical compositions, we further calculated the enhancement ratio ($ER$) of different species using BC as a proxy for primary emissions (Eq. 1):

$$ER = \frac{X_{aged}}{BC_{aged}} / \frac{X_{fresh}}{BC_{fresh}}, \qquad (1)$$

where X represents a certain species, e.g., $NO_3^-$, $SO_4^{2-}$, OA or OOA. BC was measured via a seven-wavelength Aethalometer (AE33) as described in the following section. The $ER$ >1 indicates net production of species X while the $ER$ <1 indicates net loss. Bias could be introduced by the assumption that different species have the same wall-loss rate (Hennigan et al., 2011). In addition, BC measured by AE 33 could also be biased by applying a fixed MAE value when converting the optical absorption

to the mass concentration, which instead varies with BC mixing state (Zanatta et al., 2018).

In addition, we also investigated the evolution of aerosol size distributions along with the aging process and its possible influence on estimated BrC MAE. The particle number size distribution was obtained via a Differential Mobility Analyser (DMA, Model 3082, TSI Inc., USA) combined with a Condensation Particle Counter (CPC, Model 3788, TSI Inc., USA), focusing on particles within a size range of approximately 12–460 nm.

**2.4 Optical measurement**

The light absorption of aerosols was measured with a seven-wavelength Aethalometer (Model AE33, Magee Scientific, Berkeley, CA, USA). The AE33 measures light transmitted through a filter on which particles are deposited and automatically compensates for the loading effect and multiple scattering coefficients ($C$) in real time (Drinovec et al., 2015; Drinovec et al., 2017). In this study, a newer filter tape (M8060) with a recommended $C$ value of 1.39 was used. A value of 7.77 $m^2\ g^{-1}$ was

used to convert measured absorption at 880 nm by AE33 to the mass concentration of BC (Drinovec et al., 2015). The AAE is an important parameter to characterize the spectral dependence of aerosol absorption, and was calculated via Eq. 2:

$$b_{abs}(\lambda) = K \times \lambda^{-AAE}, \qquad (2)$$

where $b_{abs}$ ($\lambda$) is the absorption coefficient at the wavelength of $\lambda$ in unit of M $m^{-1}$, and $K$ is a constant. The absorption coefficients of seven wavelengths were used to fit the exponential function curve to obtain AAE. Assuming BC is the only

light-absorbing component at 880 nm (Kirchstetter et al., 2004; Kirchstetter and Thatcher, 2012), the $b_{abs}$ of BC and BrC at 370 nm was then calculated by Eq. 3-4:

$$b_{abs,BC}(\lambda) = b_{abs}(880\ nm) \times \left(\frac{\lambda}{880}\right)^{-AAE_{BC}}, \tag{3}$$

$$b_{abs,BrC}(\lambda) = b_{abs}(\lambda) - b_{abs,BC}(\lambda), \tag{4}$$

$$MAE(\lambda) = \frac{b_{abs,BrC}(\lambda)}{OA}, \tag{5}$$

Here, $AAE_{BC}$ was assumed to be 1.1, which represents the likely range of AAE for BC externally and internally mixed with non-absorbing materials (Lack and Langridge, 2013; Li et al., 2022b; Tian et al., 2023). Uncertainties may arise from the assumption that BC is the only light-absorbing component at 880 nm. A recent study suggested that tar BrC can also exhibit significant absorption at 880 nm, with MAE ranging from 0.2 to 1.8 $m^2\ g^{-1}$ (Corbin et al., 2019). Furthermore, the use of a fixed $AAE_{BC}$ introduces additional uncertainty, as a wide range of 0.8−1.4 has been reported in previous studies (Lack and

Langridge, 2013). The MAE of BrC at different λ were further calculated based on the mass concentrations of OA (Eq. 5).

     To further quantify the contributions of different OA components to BrC absorption, a multiple linear regression (MLR) model was applied to obtain the MAE values for HOA-1, HOA-2 and OOA as follows:

$$b_{abs,BrC} = m_1 * [HOA - 1] + m_2 * [HOA - 2] + m_3 * [OOA], \tag{6}$$

Here, $m_1$, $m_2$ and $m_3$ denote the MAE, in unit of $m^2\ g^{-1}$. The light absorption and MAE in this study refer to the wavelength at

370 nm, unless otherwise noted. The $ER$ of BrC absorption was also calculated using a similar equation as for chemical components, with X representing the absorption of BrC at 370 nm and the BC term substituted by absorption at 880 nm.

## 3 Results and discussion

### 3.1 Chemical compositions of fresh and aged smoke

The chemical components of fresh and aged smoke from different biomass burning are illustrated in Fig. 1a. For fresh smoke,

the mass fraction of OA is the highest (39%–63%). The relative importance of BC and $Cl^-$ varies significantly with biomass types. For example, the median mass fractions of BC emitted by apple branch and soybean straw are relatively high, reaching 22% and 27% respectively, but are only around 7.8%–10% by other biomass types, which instead exhibit higher mass fractions of $Cl^-$ (21%–34%) and $NH_4^+$ (10%–13%). Relatively high mass fraction of $Cl^-$ (>20%) was also reported in the smoke from straw combustion (Ni et al., 2017; Ma et al., 2019). This difference may be related to local crop fertilization, which significantly

affects the element content of the straw (Huan-Cheng et al., 2005). Higher MCE may also cause more emission of Cl$^-$ due to higher burning temperatures (Wang et al., 2020b). The mass fractions of SO$_4^{2-}$ and NO$_3^-$ are very low (<5%) except for apple branch, consistent with other laboratory results (Li et al., 2016; Ma et al., 2019; Guo et al., 2022). Upon the aging process, significant changes were found for aerosols with secondary sources, including SO$_4^{2-}$, NO$_3^-$ and OA. For SO$_4^{2-}$, the *ER* is around 1.1–2.3 at A-2 and further increases to 1.7–3.1 at A-7 (Table S2). The *ER* of NO$_3^-$ at A-2 (1.5–2.9) is higher than that

of SO$_4^{2-}$, which could be explained by a faster NO$_3^-$ production rate and is also consistent with the higher NO$_x$ consumption rate compared to SO$_2$ (Fig. S3). The *ER* of NO$_3^-$ at A-7 (1.6–2.6), however, is similar to or even lower than that at A-2, and hence is significantly lower than that of SO$_4^{2-}$. Similar trends of SO$_4^{2-}$ and NO$_3^-$ from A-2 to A-7 have also been reported in other studies (Guo et al., 2022), and are attributed to the replacement of NO$_3^-$ by more acidic SO$_4^{2-}$. Similarly, Cl$^-$ depletion can also result from acid replacement by stronger acids such as H$_2$SO$_4$ and HNO$_3$ (Wang et al., 2019) or reactions with organic

acids (Laskin et al., 2012), leading to a low *ER* value of Cl$^-$ (i.e., *ER* < 1). Moreover, the cycle between NO$_x$ and their oxidative reservoir (NO$_z$), has a significant impact on the NO$_2$/NO$_z$ ratio, and the photolysis of particulate nitrate (pNO$_3$) is proposed as a potentially important mechanism influencing the partitioning between NO$_x$ and NO$_z$ (Ye et al., 2023). However, the photolysis rate constant of pNO$_3$ is highly variable and can be greatly affected by aerosol properties, including pNO$_3$ loading and particle size (Ye et al., 2017; Andersen et al., 2023). This may also help explain the observed differences in the *ER* of NO$_3^-$

between A-2 and A-7.

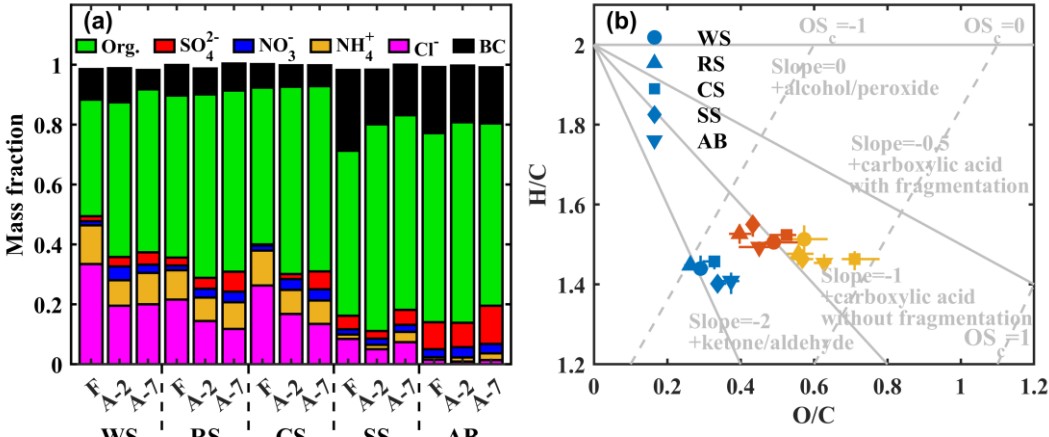

**Figure 1. (a) The mass fractions of chemical components of different biomass smoke and (b) the Van Krevelen diagram of H/C versus O/C ratios for all OA. The solid markers and whiskers in (b) denote the median, 25th and 75th percentiles, respectively, and the blue, red and orange represent fresh (F), 2-days (A-2) and 7-days (A-7), respectively. The grey dotted lines represent the estimated carbon oxidation states ($OS_c \approx 2\ O/C - H/C$), and the grey solid lines represent the evolution of OA composition.**

For OA, the *ER* increases from 1.3–1.6 at A-2 to 1.4–1.9 at A-7, showing continuous formation of secondary organic aerosols (SOA). This could be verified by the stack diagram of the high-resolution mass spectra in Fig. S4, which demonstrates a notable increase in signals at m/z 28 and 44 for all biomass types under the OH exposure, indicative of the formation of oxygenated OA, such as organic acids (Aiken et al., 2008; Lambe et al., 2013). To further describe the implied changes in OA composition, the H/C and O/C ratios for both fresh and aged smoke are distributed on the Van Krevelen diagram in Fig. 1b. The ratios are estimated based on high-resolution fragments (i.e., m/z of 43 and 44) using calibration factors from Canagaratna et al. (2015), which derived these factors from sampling standards, assuming that the elemental composition of the original species corresponds to the averaged ion composition across the mass spectrum. The H/C and O/C values of fresh OA are 1.44 and 0.33, respectively, indicating a low carbon oxidation state ($OS_c \sim -1$). Compared to fresh OA, the O/C ratios progressively increase to 0.45 for A-2 and 0.57 for A-7, accompanied by an $OS_c$ approaching 0, which is attributed to the formation of oxygenated OA. The H/C ratios also show an increase (1.52) for A-2 but slightly decrease for A-7 compared to A-2. The evolution of H/C depends on the precursor, and in general it decreases with OH exposure due to the hydrogen loss from the C=O bond formation (Lambe et al., 2013). However, Hennigan et al. (2011) also reported a slight increase in H/C ratios. Li et al. (2023b) attributed the discrepancy to the abundance of $C_2H_3O^+$ in fresh smoke, where low $C_2H_3O^+$ levels lead to an increase in m/z 43 during initial aging process.

### 3.2 OA classification based on PMF

To further explore the evolution of different OA components, we classified total OA from both fresh and aged smoke into different categories based on the PMF model. We ran the model with the number of factors ranging from 2 to 7 and chose three factors for the final results based on optimal fit and interpretability (Fig. 2). Factor 1, classified as HOA-1, has the lowest O/C ratio (0.08) and shows strong correlations with ions at m/z 55 ($r^2 = 0.85$) and 57 ($r^2 = 0.90$), respectively. Many field observations indicate that this factor is usually related to different combustion processes (Thamban et al., 2017; Rivellini et al.,

2020; Li et al., 2022a). Factor 2, classified as HOA-2, is also characterized by hydrocarbon fragments, especially at m/z of 41, 43, 55, 57, 67, 69 and 71. It has moderate ratios of O/C (0.37) and H/C (1.2) compared to other factors. It also has a good correlation with BC ($r^2 = 0.50$), ions at m/z 60 ($r^2 = 0.41$) and 73 ($r^2 = 0.68$), respectively (Table S3), which is also similar to the HOA resolved from ambient OA (Thamban et al., 2017; Li et al., 2017; Wang et al., 2020a). Factor 3 is dominated by ions at m/z 28 and 44 with the highest O/C ratio of 0.76. Moreover, it is well correlated with oxidized products or fragments, including $SO_4^{2-}$ ($r^2 = 0.51$), $NO_3^-$ ($r^2 = 0.58$), m/z 28 ($r^2 = 0.72$) and m/z 44 ($r^2 = 0.90$), and thus is classified as OOA.

For fresh smoke, the OA is dominated by HOA-1 and HOA-2, while the contribution of OOA is still of significance (19%–40%, Fig. 2d). This is actually a significant feature distinguishing biomass sources from other sources (Aiken et al., 2008). Biomass is naturally rich in oxygen-containing compounds, such as cellulose, hemicellulose and lignin. And those oxygen-containing structures partially break down and release oxygen-rich substances into the gas or aerosol phase during combustion. Incomplete combustion during ignition and the burnout stages can also produce oxidized OA with relatively high O/C ratios (Heringa et al., 2011; Avery et al., 2023). Previous studies have reported O/C ratios of 0.2–0.6 in fresh biomass burning smoke (Heringa et al., 2011; Fang et al., 2017; Ma et al., 2019; Li et al., 2023b), significantly higher than those from traffic exhaust (around 0.02–0.19) (Chirico et al., 2010; Dallmann et al., 2014; Collier et al., 2015). Along with OH exposure, the fractions of HOA-1 and HOA-2 decrease while OOA increases significantly, reaching 55%–70% at A-2 and 62%–80% at A-7, respectively. For HOA-1, the *ER* for all biomass types decreases from 0.88 at A-2 to 0.66 at A-7, respectively. The *ER* of HOA-2 is around 0.63 at A-2, but slightly increases to 0.71 at A-7. The *ER* of OOA of different biomass types reaches 2.6–3.7 at A-2, implying significant and fast yields of SOA. The *ER* of OOA continues to grow at A-7 but the growth rate slows down compared to A-2. Previous studies have also shown that the initial rapid increase of SOA and O/C in biomass burning aerosols slows down with aging time exceeds 2 days due to the depletion of precursors (Grieshop et al., 2009; Cappa et al., 2020; Li et al., 2024). It is worth mentioning that there exists a large variability in OA composition as well as its evolution upon aging among different biomass types. For instance, the *ER* of OOA in AB smoke is the smallest and the difference between A-2 and A-7 is minor, while the *ER* of OOA in wheat straw smoke increased by nearly 63% from A-2 to A-7.

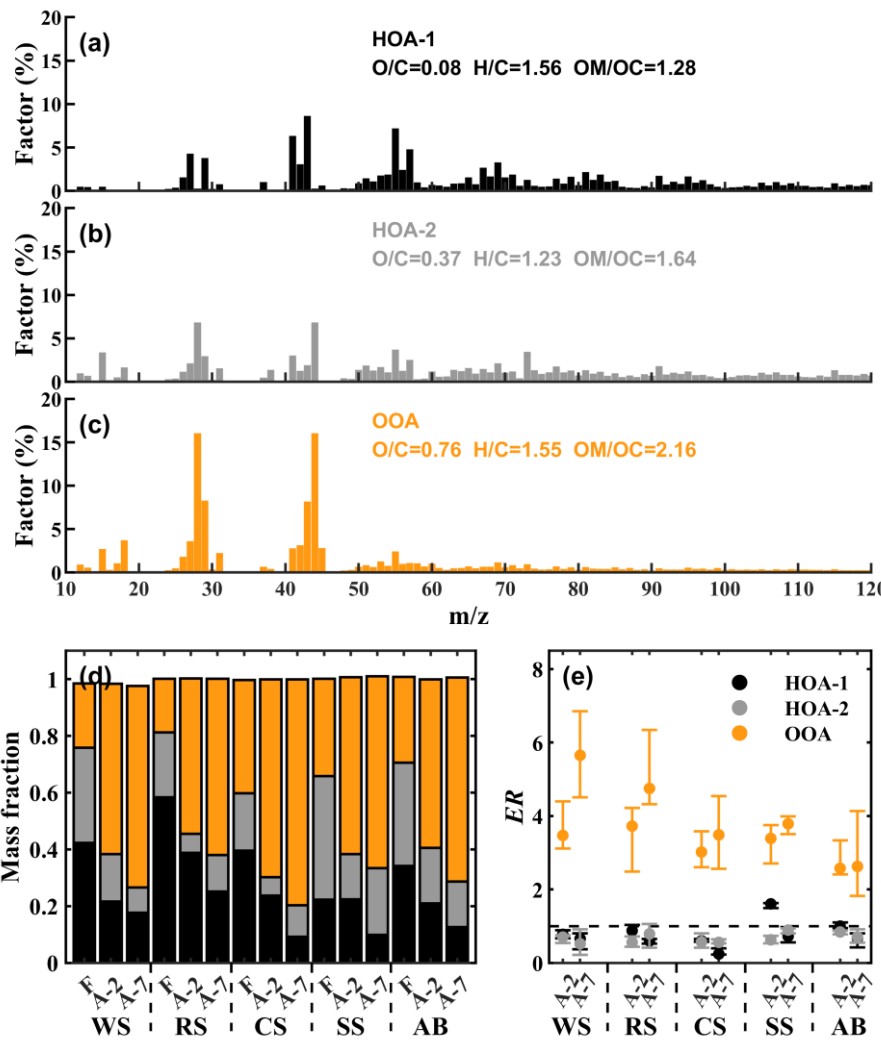

**Figure 2. (a-c) High-resolution mass spectra of the three OA factors. (d) The median mass fractions and (e) the enhancement ratio (*ER*) of different OA factors at different aging levels. The solid circles and whiskers in (e) denote the median, 25th and 75th percentiles, respectively.**

## 3.3 Evolution of particle size

The particle number size distribution of freshly emitted biomass smoke exhibits a unimodal lognormal distribution. After PAM aging, it exhibits a bimodal pattern, with an additional small-particle mode (around 20 nm) compared to fresh smoke, mainly resulting from nucleation. Here we only focus on the evolution of particles in accumulation mode. The maximum peak diameter

($D_m$) and half-peak widths ($\sigma$) of Gaussian fits are shown in Fig. 3. The wheat straw shows the largest $D_m$ (239 nm) of freshly emitted biomass aerosols, followed by corn straw (233 nm), soybean straw (204 nm), rice straw (178 nm) and apple branch (162 nm). The $D_m$ is within the range (50–500 nm) reported by Chen et al. (2019). Previous studies also show that the wheat smoke has a higher $D_m$ in residue crops (Fang et al., 2017; Chen et al., 2019). Variations in $D_m$ is related to both biomass types and combustion conditions (Park et al., 2013; Ma et al., 2019), such as fuel to air ratio, moisture content and density. Furthermore, it can be seen that rice straw has the smallest $\sigma$ (0.22), while apple branch has the largest $\sigma$ (0.28) and error bar (Fig. 3b), indicating that the particle number size distribution of apple branch is the widest, while that of rice straw is the narrowest.

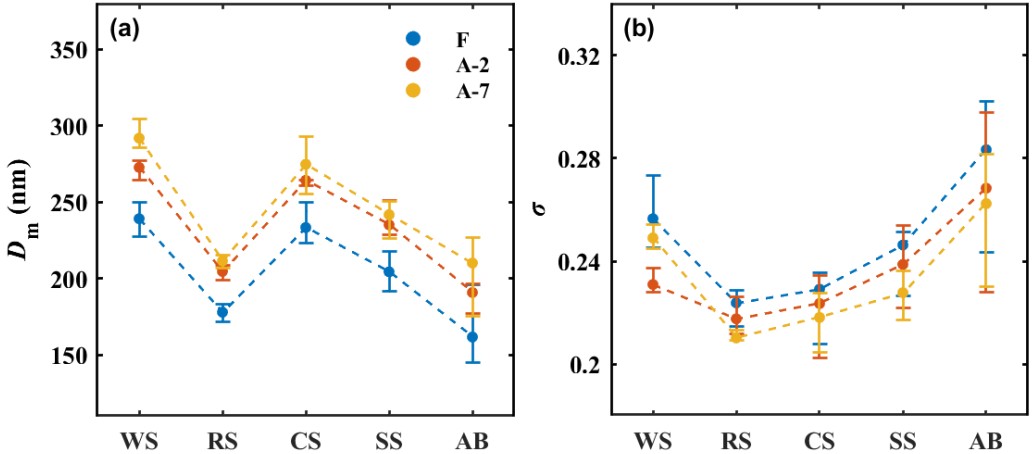

**Figure 3. (a) The maximum peak diameter ($D_m$) and (b) half-peak widths ($\sigma$) from Gaussian fits for particle number size distribution. The solid circles and whiskers denote the median, 25th and 75th percentiles, respectively.**

A significant increase in $D_m$ is observed with increasing OH exposure, about 15% and 17% increase for A-2 and A-7, respectively. Similar change has also been reported in previous studies. For instance, Ma et al. (2019) reported a 1.1–1.4 times increase in particle size during 4-hour aging and Zhao et al. (2022) showed 1.2 times increase after 15-day photooxidation. The increase for 7-day is only slightly higher than that for 2-day, indicating a significant slowdown in particle growth. Many studies have also shown that $D_m$ in biomass burning aerosols increases slowly after a rapid increase upon OH exposure (Fang et al., 2017; Reyes et al., 2019). This could be explained, on one hand, by the limited availability of precursors. On the other

hand, as particles grow, more mass is required to increase their diameter by one unit. Consequently, smaller articles grow faster in diameter, leading to a narrower size distribution and a reduced $\sigma$ upon aging (Fig. 3b).

## 3.4 Light absorption properties

A significant BrC contribution to the total light absorption in fresh smoke is observed but with a large variability across different biomass types ($f_{BrC}$ = 29%–60%, Fig. 4). These values are in the middle range of the results in previous studies (~10%–
90%) (Tian et al., 2019; Zhang et al., 2020; Fang et al., 2022). The presence of BrC makes the AAE of total aerosol significantly deviated from 1, with the highest value for rice straw (2.3), followed by corn straw (1.8), apple branch (1.6), wheat straw (1.5) and soybean straw (1.4). These AAE values are also within the range of previous studies for biomass burning aerosols (Laskin et al., 2015). Under OH exposure, the AAE decreases due to a reduction in BrC light absorption, with the $ER$ of $b_{abs, BrC}$ around 0.51–0.85 and 0.63–0.90 in A-2 and A-7, respectively. The difference is statistically significant with $p < 0.05$, except for rice
samples. The large decrease in BrC absorption at A-2 implies a significant effect of photobleaching associated with OH-driven oxidation in the PAM. No significant difference is observed in the $ER$ of BrC absorption between A-2 and A-7, indicating a limited photobleaching effect after 2-day aging. It has been reported that 20%–64% of BrC absorption remains even after a longer aging level (Sumlin et al., 2017; Browne et al., 2019; Hems et al., 2021), consistent with our results. Additionally, an exponential relationship between AAE and $f_{BrC}$ is clearly observed (AAE = 0.97*exp (1.4*$f_{BrC}$), with 95% confidence intervals
of 0.93–1.02 and 1.29–1.52 for the former and latter coefficients, respectively). Similar relationship has also been reported by Sun et al. (2021) for fresh household biomass burning smoke, with AAE = 1.01*exp (1.8*$f_{BrC}$), which falls within the uncertainty range of our results. And our results further demonstrate that the relationship stands for both fresh and aged biomass burning emissions.

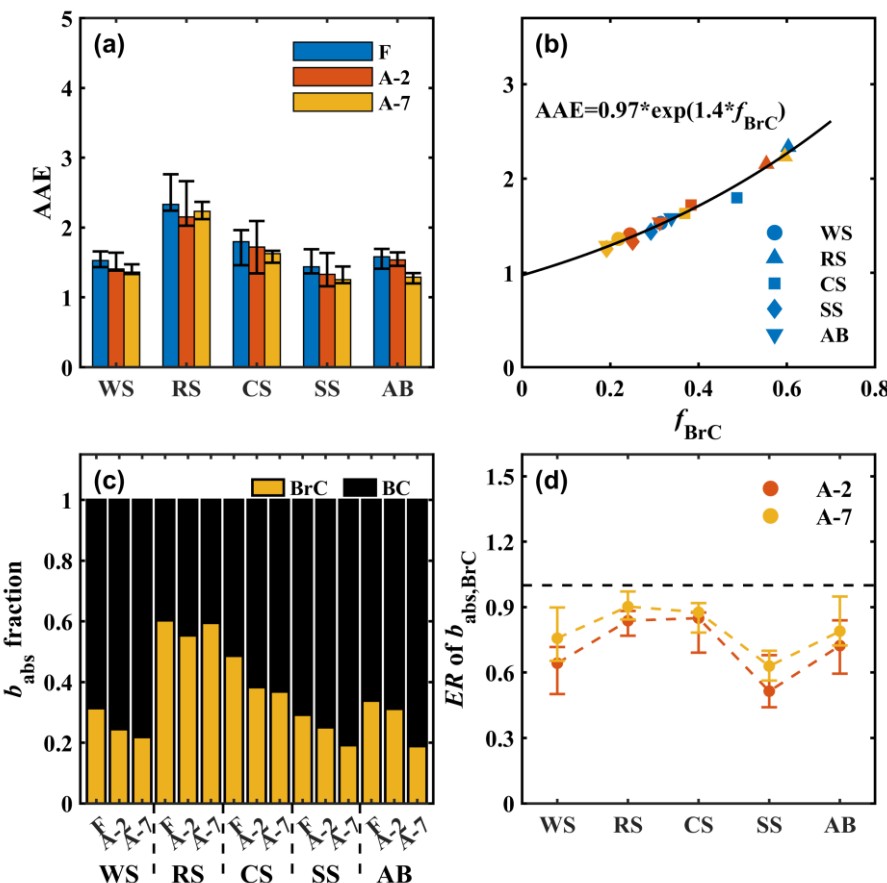

**Figure 4. (a) AAE of total aerosol and (b) its relationship with BrC contribution to the total light absorption ($f_{BrC}$) at 370 nm; (c) relative contribution of BrC and BC to total light absorption at 370 nm; and (d) the enhancement ratio ($ER$) of BrC absorption.**

In addition to the variation in BrC absorption, the MAE of fresh OA emitted from different biomass burning also exhibits a large range, with the highest value from rice straw (5.7 m$^2$ g$^{-1}$), followed by apple branch (3.9 m$^2$ g$^{-1}$), soybean straw (3.7 m$^2$ g$^{-1}$), corn straw (3.5 m$^2$ g$^{-1}$) and wheat straw (2.1 m$^2$ g$^{-1}$) (Fig. 5a). Upon OH exposure, a large decrease in the MAE is observed, reduced by 32%–64% and 37%–68% at A-2 and A-7, respectively, compared with fresh. The change in MAE is associated with the changes in both BrC absorption and OA mass concentrations. In other words, it is due to the combined effect of bleaching and the formation of SOA with weak absorption. Compared to fresh smoke, the $b_{abs, BrC}$ at A-2 is reduced by about 15%–49%, accounting for roughly half or more of the change in MAE. This suggests that the reduction in MAE at A-2 is dominated by both $b_{abs, BrC}$ and OA mass. Moreover, the MAE of rice straw, corn straw and soybean straw at A-7 is similar to

that in A-2, which may imply the resistance of BrC to photobleaching after the first few days of aging. Similarly, Zhao et al.

(2022) also reported that 49% to 67% of the initial MAE of rice straw remained after equivalent 9 days of aging.

    To further distinguish the light absorptivity of different OA components, we calculated the MAE of the three OA factors based the MLR method (Fig. 5b). There exists a significant difference in the MAE of the three OA factors, with the highest value (5.6 $m^2$ $g^{-1}$) for HOA-1, followed by HOA-2 (4.0 $m^2$ $g^{-1}$) and OOA (0.76 $m^2$ $g^{-1}$). The decrease in the MAE with the

increase in O/C ratio is consistent with previous findings (Sumlin et al., 2017; Schnitzler et al., 2020; He et al., 2022). As discussed above, the evolutions of three OA factors are quite different during the aging process, their contribution to the light absorption thus varies significantly. For A-2 and A-7, the $ER$ of $b_{abs, \, BrC}$ for HOA-1 are 0.85 and 0.60, respectively, and HOA-2 shows similar values of 0.69 and 0.79. Contrarily, OOA exhibits the highest $ER$ of $b_{abs, \, BrC}$ (3.2 and 4.0, respectively). Consequently, the contribution from OOA absorption increases from 6% at fresh to 19% at A-2 and 26% at A-7, respectively,

while the contribution from HOA-1 absorption decreases from 65% at fresh to 59% at A-2 and 43% at A-7, respectively, and that from HOA-2 remains similar (23%–30%) (Fig. S6). The significantly different behavior of the three OA factors combined with their distinct MAE values suggest the importance in climate models to classify OA into different groups based on their optical properties, which could be represented by the oxidation sate in this case.

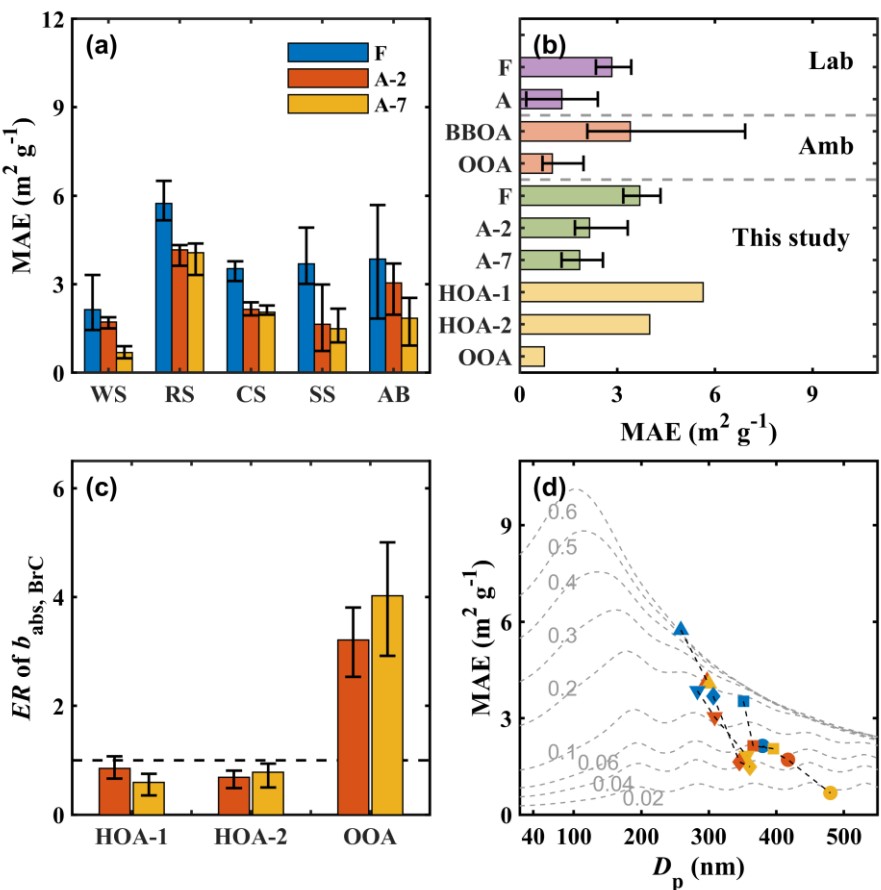

**Figure 5. (a) MAE of BrC at 370 nm and (b) comparison of BrC MAE in this study with previous studies including both laboratory measurements and field observations (see details and corresponding references in Table S4 and S5); (c) the enhancement ratio (*ER*) of BrC absorption in different OA factors; (d) relationship of MAE at 370 nm with particle diameter ($D_p$), the blue, red and orange dots represent the peak sizes of the particle mass size distribution from fresh (F), 2-days (A-2) and 7-days (A-7), respectively. Note that the MAE of OC in some studies has been converted to the MAE of OA assuming OA/OC equal to 1.8; the MAE dependence on**
**particle size in (d) is calculated using the Mie model for pure BrC (gray dashed lines).**

Additionally, we found that the MAE of OA decreases with the increases of particle size (Fig. 5d). To explore the potential influence of particle size on the variation in BrC MAE, we conducted a theoretical calculation of the MAE for pure BrC particles as a function of $D_p$ using Mie theory. Assuming that the BrC particles are spherical with as a density of 1.6 g cm$^{-3}$, the MAE for BrC particles at a specific diameter can be determined, given a particular refractive index. In this study, we adopt

the real part (*n*) of the refractive index of pure BrC is 1.7 from Saleh et al. (2014) and the *k* varies from 0.6 to 0.02. As shown

in Fig. 5d, for a fixed particle diameter, the BrC MAE increases with $k$. The relationship between particle diameter and MAE is more complicated even with the refractive index remains constant. For small particles (<100 nm), the MAE generally increases with diameter (gray dashed lines). However, for large particles, the MAE decreases with diameter given a larger $k$ value but exhibits weak fluctuations and no obvious trends when $k$ is small, which is consistent with previous studies (Hems et al., 2021). The median diameters of BrC particles in both fresh and aged biomass burning plume detected in this study are in the region where MAE either decreases with diameter or shows minimal dependence on diameter (Fig. 5d). Therefore, the large reduction in MAE is mainly driven by changes in $k$, specifically, lower $k$ values at larger $D_\mathrm{p}$. As discussed earlier, upon OH exposure, there is a significant increase in particle size with new components formation (Fig. 3). The latter is dominated by more contribution from OOA with high O/C (Fig. S5b), which has much lower MAE or $k$ compared to the other two OA factors.

For better comparison, Fig. 5b also summarizes MAE values for biomass burning OA reported from both laboratory measurements and field observations. The MAE of fresh BrC either measured in the laboratory or derived from ambient observations shows a large variability but has a similar range as with our results. This large variability is primarily attributed to differences in biomass types and burning conditions. As mentioned above, the MAE or $k$ of biomass burning OA could be expressed as a function of BC/OA, accounting for the influence of burning conditions (Saleh et al., 2014; Pokhrel et al., 2016). Here, we found similar relationship with a linear regression slope of 8.0 (Fig. S5a). Xie et al. (2017a) and Park et al. (2020) also reported a slope of approximately 0.63 and 28, respectively, for biomass burning smoke. The large differences in the slope may be due to variations in the biomass type and combustion conditions. The MAE for aged OA is still very limited in laboratory measurements, with a value around 1.3 $\mathrm{m^2\ g^{-1}}$ that largely depends on the simulated aging levels. This is also consistent with our results, 2.1 and 1.8 $\mathrm{m^2\ g^{-1}}$ for OA at A-2 and A-7, respectively. The MAE of SOA derived from ambient observations is slight lower. This is because SOA from ambient studies includes precursors from sources other than biomass burning. It has been shown that SOA from biogenic sources is less light-absorbing compared to that from anthropogenic sources, including biomass burning (Du et al., 2014).

**4 Conclusions**

The optical, physical and chemical properties of the biomass burning smoke upon aging were monitored simultaneously to better understand the evolution of biomass burning aerosol, especially BrC, and its driving factors. Upon OH exposure, the fraction of secondary components increases significantly, and the increase in OA is associated with the rising value of O/C. However, the behavior of the secondary components differs significantly at different aging levels. The $ER$ of $NO_3^-$ at A-2 is similar to or lower than A-7, probably due to the replacement of $NO_3^-$ by more acidic $SO_4^{2-}$ or the photolysis of particulate

nitrate. The $ER$ of OA at A-2 and A-7 are also very similar due to limited availability of precursors. The particle number size distribution of fresh biomass smoke exhibits a unimodal lognormal distribution. Particle growth is mainly dominated by the formation of SOA and thus shows a similar trend as OA upon aging.

    The optical properties of the biomass burning obtained in this study are in general within the wide range reported in previous studies, including the relationship between the AAE of total aerosol and the fraction of BrC absorption, the dependence of BrC

MAE on the BC/OA ratio. The large variation in the quantified relationship, however, emphasizes the significant influence from the biomass types and warrants more studies. The extent of the bleaching as well as the formation of SOA with weak absorptivity together determine the evolution of BrC MAE at different aging levels. There exists a large decrease in BrC MAE from fresh smoke to A-2, contributed comparably by both the bleaching in BrC absorption and the increase in SOA. We also observed a negative correlation between the MAE and the particle size. However, based on the Mie theory calculation, we

found that the change in MAE is mainly driven by the change in $k$ of OA instead of the particle size, implying lower $k$ value at larger particle size. This is consistent with the formation of SOA with lower MAE (i.e. $k$) upon aging, which dominates the particle growth. Therefore, it is important to distinguish the behavior of different OA components and their contribution to BrC absorption.

    The study further classified the OA into three factors based on the PMF model, among which, HOA-1 and HOA-2 are more

related to fresh smoke while OOA is associated with secondary formation with higher O/C value. The MAE of different OA factors also differ from each other, decreasing as the O/C value increases. The behavior of different OA factors upon aging also shows distinct patterns, with a significant decrease in HOA-1 and HOA-2 but an increase in OOA. For A-2 and A-7, the $ER$ of $b_{abs, BrC}$ for HOA-1 are 0.85 and 0.60, respectively, and HOA-2 shows similar values of 0.69 and 0.79. Contrarily, OOA

exibits the highest *ER* of $b_{abs, BrC}$ (3.2 and 4.0, respectively). Our results thus demonstrate the necessity of classifying OA into different categories based on their distinct MAE and behavior upon aging. Future studies should focus on the evolution of different OA groups rather than the whole OA, which could be classified by their O/C value, solubility, etc., to develop more appropriate BrC parameterization in model studies for better assessing its climate effects.

**Code and data availability.** The codes used to curate and analyze the datasets and produce the figures and results of the study are available from the corresponding author upon request. The data supporting the conclusions of this paper are available from https://doi.org/10.5281/zenodo.15493841.

**Author contributions.** QQW, QYW, NM, JT and YQZ conceptualized and designed the study. NM, QYW and YFW provided instrumentation and experimental materials. ZY, GX, MG, XXZ, YZ and WKR performed the laboratory measurements. ZY completed the formal data analysis and prepared the original manuscript draft with the help of QQW, NM, QYW, JCT, JH, JT, NY, YQZ, JJC, HS and YFC. All the authors reviewed, edited, and contributed to the scientific discussion in the manuscript.

**Competing interests.** The authors declare that they have no conflict of interest. Some authors are members of the editorial board of journal X.

**Acknowledgements.** We gratefully acknowledge the Institute of Earth Environment of Chinese Academy of Sciences (IEECAS) for the provision of the experimental site, equipment and technology. The authors gratefully thank the people at all sites for sample collections and all of the individuals and groups that participated in this project.

**Financial support.** The work was supported by the special fund of National Natural Science Foundation of China (grant nos. 42377093 and 42375072), Guangdong Basic and Applied Basic Research Foundation (grant no. 2024B1515040026), the National R&D Program of China (grant no. 2024YFC3712900), the National Key R&D Program of China (grant no. 2022YFF0802501), the "Western Light"–Key Laboratory Cooperative Research Cross-Team Project of Chinese Academy of Sciences (grant no. xbzg-zdsys-202219), the Guangdong Innovative and Entrepreneurial Research Team Program (Research Team on Atmospheric Environmental Roles and Effects of Carbonaceous Species: grant no. 2016ZT06N263), and Special Fund Project for Science and Technology Innovation Strategy of Guangdong Province (grant no. 2019B121205004).

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
