# Peer review of "Laboratory studies on the optical, physical, and chemical properties of fresh and aged biomass burning aerosols"

_EGUsphere, 2025_

## Author Comment (AC1)

**Reviewer #1**

**General Comments:**

The authors have addressed all of my previous comments. I recommend the paper for publication in its current form.

Thank you for your previous suggestions, which are very helpful to improve the coherence, clarity and scientific rigor of the manuscript. Thank you again for your support to us.

---

## Author Comment (AC2)

**We thank the reviewers for their supportive and thoughtful comments. We have addressed all concerns raised by the reviewers. Please see our point-by-point response to the comments below, highlighted in blue.**

**Reviewer #2**

**General Comments:**

The authors present here results of fast photochemical aging of particles and gases derived from combustion of various fuels, mainly straws. The measurements appear to be of good quality. Their work adds to the literature, but could make much stronger connections to the existing literature to allow for stronger conclusions. This is especially the case when it comes to thinking about and discussing other laboratory aging experiments in the literature; I find the author's approach to be fairly selective and a bit unfocused. There are also quite a few statements and conclusions made that require more support or context if they are going to stand. I also strongly encourage that the data underpinning this work be made available in an open, doi-referenced archive.

We thank the reviewer for their supportive and thoughtful suggestions which are of great value for improving the quality of our paper. We have made the corresponding revisions based on the comments, including a more in-depth discussion to better support the conclusions, more accurate citation of relevant literature, and other necessary adjustments. Additionally, the data supporting this work has been made publicly available in an open-access, doi-referenced archive.

**Specific comments:**

L44: Here, the authors state that positive forcing can reach "up to +0.05 W/m$^2$." Yet, on line 32 they state that the forcing may be up to +0.57 W/m$^2$. It would be good to have some consistency.

Sorry for the confusion. The range of 0.03-0.57 W/m$^2$ refers to the warming effect of BrC due to its light absorption while "up to 0.05 W/m$^2$" means the radiative forcing effect of biomass burning aerosols, which includes the cooling effect of organic aerosols due to the light scattering. This sentence here is mainly intended to convey that, when considering BrC light absorption, the overall radiative forcing of biomass burning aerosols can shift from a cooling effect to a warming effect. To avoid confusion, we have revised the sentence as: "... When considering BrC light absorption, the overall regional or global radiative forcing of biomass burning aerosol can shift from negative to positive effects (Saleh et al., 2015). …"

L47: The choice of Huo et al. (2018) seems random. Why this study? Especially given that it focuses on HULIS.

We agree with the reviewer that this reference is only one of many studies examining the influence of combustion conditions on the optical properties of BrC. It specifically discusses how fuel moisture content or stacking conditions affects combustion

efficiency, which in turn affects HULIS formation and its optical properties. To avoid potential confusion and to improve the clarity and coherence of the manuscript, we revise the sentences into: "... The optical properties of freshly emitted BrC, such as the absorption Ångström exponent (AAE) and mass absorption efficiency (MAE) or imaginary part ($k$), have been shown to strongly depend on burning conditions, with BrC from flaming biomass burning combustion exhibiting higher MAE and lower AAE than that from lower-temperature smoldering combustion (Laskin et al., 2015; Saleh, 2020). …"

L50: It would be helpful to clarify/distinguish between process-level studies and ambient studies. Park et al. is an ambient study, and thus divining relationships is more complicated than in process-level studies such as Saleh and Xie.

Thanks for the comment. We agree with the reviewer that ambient studies are more complicated, and relationships are generally developed based on laboratory studies. The references we cite are all based on laboratory studies, including Park et al. (2020), which tested four biomass types in a laboratory combustion environment. To avoid confusion, we revise the sentence into: "… Utilizing the ratio of BC/OA as a proxy of the burning condition, several studies have established a quantitative relationship between the MAE and the burning conditions **based on laboratory combustion experiments** (Saleh et al., 2014; Xie et al., 2017; Park et al., 2020)."

L61: I encourage the authors to clarify what "formation of nitroaromatic compounds" means more specifically. Is this SOA formation? Or are they talking about heterogeneous transformations. This matters to thinking about process. Moreover, the paper cited her is about photooxidation of naphthalene and SOA formation, while the previous sentence implies an importance of "nitration of existing OA." These are not at all the same thing, leading to ambiguity and some contradictions in the authors' presentation of the literature.

Sorry for the confusion. We now revised the two sentences to: "... Moreover, the light absorption of BrC can be enhanced through the functionalization and polymerization of existing OA (Wong et al., 2019; Hems et al., 2020) or the formation of new nitrogen-containing organic compounds, e.g., nitroaromatics (He et al., 2022; Yang et al., 2022). …"

L63: Here, I find that the authors continue to mix and match studies without really focusing on what people did in a sufficiently careful manner. The cited Choudary study is specifically about photooxidation of collected biomass burning particles in aqueous solutions while the Schnitzler study is of suspended particles produced from smoldering in the lab and then the Fang study is also for suspended particles in a chamber. Are these comparable? Maybe? But the authors don't give the context necessary for a reader to make the connections. I encourage the authors to be more precise throughout their comparisons between studies in the introduction. The way things are presented seems a bit random to me.

We agree with the reviewer that there are differences in the experimental conditions of

these studies. The main purpose here is to illustrate the complexity of the BrC aging process, as shown by existing studies. Since the Choudary study mainly dealt with aqueous-phase changes, we now substituted it with Liu et al. (2021) study, which also focuses on suspended particles to enhance comparability. The sentences now are revised to: "... The photobleaching and photo-enhancement may occur concurrently, making the evolution of BrC absorptivity more complicated. **Some studies reported continuously decrease in $k$ (Liu et al., 2021) while some show a slight increase first followed by a significant decrease (Cappa et al., 2020; Schnitzler et al., 2020)**. …"

L65: The authors here allude to "only a few studies" but fail to cite any. Which studies?

Thank you for the suggestion. We now revised the sentences: "... However, current studies mainly focus on the evolution of the overall BrC absorptivity and few have endeavored to distinguish the behaviors of different BrC components (Wong et al., 2019; Fleming et al., 2020), which may undergo totally different aging processes. …"

L87: Please indicate which lights (wavelength) were used in the PAM. Also, more details of the operation are needed. The authors simply state that the lights were turned on for 40 minutes. However, the intensity was presumably varied during this 40 minutes period, leading to changes in the OH exposure. How many conditions were considered? How long per condition? This needs to be in the main text, not the supplemental.

Sorry for the confusion. The 185-nm UV light was maintained constant for each simulated aging level. In this study, we simulated 2 aging level (2 and 7days), corresponding to two UV intensities. The residence time within the PAM is set to 90 s for all experimental conditions. To make it clear, we now added more details: "... Briefly, inside the PAM-OFR, OH radicals were formed through a series of photochemical reactions of $H_2O$ and $O_2$ under 185 nm UV illumination. The simulated OH concentrations can thus be controlled via adjusting the UV light intensity. In this study, two aging levels were simulated by applying two distinct UV intensities. For each aging level, the UV intensity was kept constant throughout the experiment by regulating the lamp voltage. …"

Table S1: Are these average conditions? If so, need to indicate. What is the reproducibility?

Yes, these are average conditions for each biomass type at a certain aging level. We have now updated the table to include the standard deviation.

Table S1. Statistics of related parameters of different biomass combustion (mean± σ).

| Biomass type | Sources | Aged day | # of tests | RH (%) | MCE |
|---|---|---|---|---|---|
| Wheat straw (WS) | Xi'an | 0, 2 | 8 | 19.9±5.0 | 0.964±0.008 |
| | | 0, 7 | 7 | 11.5±3.3 | 0.961±0.014 |
| Rice straw (RS) | Shichuan | 0, 2 | 10 | 20.0±4.8 | 0.976±0.008 |
| | | 0, 7 | 8 | 17.7±4.6 | 0.970±0.015 |
| Corn straw (CS) | Jiangsu | 0, 2 | 7 | 14.8±1.9 | 0.974±0.006 |
| | | 0, 7 | 7 | 14.8±2.3 | 0.982±0.011 |

| | | | | | |
|---|---|---|---|---|---|
| Soybean straw (SS) | Liaoning | 0, 2 | 9 | 16.6±2.5 | 0.957±0.013 |
| | | 0, 7 | 7 | 18.4±1.8 | 0.969±0.011 |
| Apple branch (AB) | Hubei | 0, 2 | 8 | 18.3±5.7 | 0.963±0.023 |
| | | 0, 7 | 6 | 16.7±1.9 | 0.969±0.009 |

L103: The authors state "Actually, flaming and smoldering phases occur simultaneously. during a fire." Why is this important to note? Some context would be helpful.

Sorry for the confusion. The sentence is mainly intended to convey that, "the experiment in this study was designed for the flaming combustion condition; nevertheless, transition period of smoldering were inevitable during ignition and the burnout stages." To avoid confusion, we now deleted it.

L106: The lifetimes of NOx and SO2 are very, very different. Yet, the loss through the OFR is similar for both. How can this be justified? Moreover, the SO2 consumption increases with aging in many cases. This seems non-physical. What is the explanation?

We agree with the reviewer that the lifetime of $NO_x$ and $SO_2$ are different, with $NO_x$ being more active than $SO_2$. This is evident in Fig. S3, where we can see the ratio of $NO_x$ measured after PAM to those before PAM is in general lower than that of $SO_2$ at the same aging level. The difference is most significant in the corn straw (CS) and apple branch (AB) cases, where the ratio of $NO_x$ is about one-third to one-half of the ratio for $SO_2$ at A-2, suggesting higher consumption of $NO_x$ than $SO_2$. It is worth mentioning that in addition to the reactions with OH, VOCs enriched in the smoke can also react with $NO_x$ and $SO_2$. Moreover, the $NO_x$ concentrations can also be re-generated by the photolysis of $HNO_3$ and nitrate on aerosol surfaces (Ye et al., 2023). This may to some extent explain the variation in the observed $NO_x$ loss relative to $SO_2$ loss and why this difference is smaller than the difference in their reaction rates with OH. An increase in $SO_2$ consumption with aging is to be expected, given the positive correlation between its consumption and the duration of aging.

L111: DeCarlo et al. (2006) do not provide a description of the ACSM. Please select an appropriate reference.

Thank you for the correction. DeCarlo et al. (2006) provide a description of the AMS rather than ACSM, we now cite the right one: " ... A detailed description of this instrument can be found in Fröhlich et al. (2013) and Xu et al. (2017). …"

L114: It is stated that "the collection efficiency (CE) value was 0.5 in this study." How was this determined? The particles are, presumably, primarily organic and often OA has a CE closer to unity, although this can change with aging. Did the authors aim to quantify this value? Or is this simply an assumption that was not tested? The authors have size distributions and so should be able to test this, with an assumption of the particle density. Is the ratio between the AMS mass and the derived particle mass from the size-distribution measurements independent of all conditions?

The value of 0.5 is adopted from the study of Middlebrook et al. (2012), which suggests a default CE vaule of 0.5 for a standard vaporizer. We now revised the text to add the

reference. In addition, we used laboratory data from SMPS and ACSM to calculate the mass concentration of particles. As shown in Figure R1, the mass concentrations from the ACSM measurements are slightly lower than those from SMPS, but the correlation between the two is very good. Bias may be introduced in the mass concentrations calculated from SMPS data due to the assumption that particles are spherical and have a constant density.

[Figure]

Figure R1. Scatter plot of the $PM_1$ mass concentration calculated by particle number size distribution from SMPS, assuming a density of 1.4 g/m$^3$, versus NR-PM from ACSM plus BC mass concentration from AE33.

L139: The authors should clarify that the MAE from Drinovec et al. is not the true MAC for BC at 880 nm. It is specific to the Aethelometer. The actual BC MAC is much smaller than the value given here. As stated, a reader is left thinking that this is the actual MAC for BC at 880 nm, which it is not.

Sorry for the confusion. As suggested, we revised the text to: "…. A value of 7.77 m$^2$ g$^{-1}$ was used to convert measured absorption at 880 nm by AE33 to the mass concentration of BC (Drinovec et al., 2015). …"

L144: The authors use a few references to indicate that BC is the only absorbing component at 880 nm. However, this is not necessarily true. BrC may be very weakly absorbing, but if the concentration of BrC is >> than BC it can still be important. What matters is the ratio [BC]*MAC_BC/[BrC]*MAC_BrC. BrC can absolutely absorb at 880 nm, although such absorption is typically weak. I encourage the authors to clarify. (In this study, OA/BC is unlikely high enough to matter, but the point is still valid.)

We agree with the reviewer that BrC may contribute to light absorption at this wavelength. We now add discussion on the uncertainty introduced by BrC absorption at this wavelength: "… Uncertainties may arise from the assumption that BC is the only light-absorbing component at 880 nm. A recent study suggested that tar BrC can also exhibit significant absorption at 880 nm, with MAE ranging from 0.2 to 1.8 m$^2$ g$^{-1}$ (Corbin et al., 2019). Furthermore, the use of a fixed $AAE_{BC}$ introduces additional uncertainty, as a wide range of 0.8−1.4 has been reported in previous studies (Lack and Langridge, 2013). …"

L157: Why focus on 370 nm? Why not take a more holistic approach to the analysis and consider the BrC across the solar spectrum, or at least at the peak?

Thanks for the suggestion. This study mainly focuses on BrC aging and therefore only one wavelength is selected to characterize its variation. BrC shows the strongest absorption at 370 nm in AE33, and this wavelength is commonly reported in the literature, which facilitates comparison with previous studies.

L162: I strongly encourage the authors to use words, rather than abbreviations here. I can't keep track of what AB and SS are, for example. Apple Branch and Soybean Straw are much more descriptive. If there were only two to keep track of that would be perhaps okay, but here there are 6 fuel types and it is easy to get lost.

Thanks for the suggestion. We now use words instead of abbreviations throughout the manuscript.

L165: The statement about how crop fertilization practices impact straw composition would be strengthened with a reference.

Thanks for the suggestion. We now add one reference to support the statement: "... **This difference may be related to local crop fertilization, which significantly affect the element content of the straw (Huan-Cheng et al., 2005)**. …"

L165: The authors state "Higher MCE may also cause more emission of Cl$^-$." But, is any relationship observed here? Is this relevant to the current study?

Sorry for the confusion. This is not the results from this study. We just cite the result from the study of xx for comparison/discussion. We now revise the content into: "… Relatively high mass fraction of Cl$^-$ (>20%) was also reported in the smoke from straw combustion (Ni et al., 2017; Ma et al., 2019). This difference may be related to local crop fertilization, which significantly affects the element content of the straw (Huan-Cheng et al., 2005). Higher MCE may also cause more emission of Cl$^-$ due to higher burning temperatures (Wang et al., 2020). …"

L167: The authors state that "Upon the aging process, significant changes were found for aerosols with secondary sources, including SO$_4^{2-}$, NO$_3^-$ and OA." However, the authors only present relative composition plots. This means that something could go down simply because something else went up more, not because it didn't go up too. It might be useful to present (in the supplemental) some aspect of changes in absolute concentration changes.

We agree with the reviewer that in a decrease in relative contribution does not necessarily mean its absolute concentration goes down. Indeed, we did calculate the enhancement ratio (*ER)* of different chemical components to show the changes in absolute concentration in Table S2.

L169: I find this argument very weak. The authors state that the higher enhancement for nitrate over sulfate could be "explained by a faster NO$_3^-$ production rate and is also consistent with the higher NOx consumption rate compared to SO2." A look at Fig. S3

indicates that, while this is technically true, the differences are small. And, moreover, inconsistent with expected OH-driven oxidation rates. Here, I feel that the authors are forwarding a narrative that is not supported by the data. I would welcome a more quantitative approach to this conclusion. Also, there is sufficient discussion of enhancement ratios that it seems appropriate that the table, or an equivalent figure, should be included in the main text. By having it in the supplemental it is too easy for details to be ignored. For example, with the NO3− production, the authors talk in broad terms but they do not ever discuss the fact that for WS the NO3- ER decreases a lot from A2 to A7 while for all other fuels it either remains essentially constant or increases slightly.

Thanks for the suggestion. As replied to the earlier comment, the reactions in PAM are quite complicated. Specifically, the $NO_x$ concentrations can also be re-generated by the photolysis of $HNO_3$ and nitrate on aerosol surfaces ($pNO_3$), which helps explain the relatively smaller difference in the depletion of $NO_x$ and $SO_2$.

As mentioned in the text (Line181), the relatively small change in the $ER$ of $NO_3^-$ from A-2 to A-7 can be attributed to the replacement of $NO_3^-$ by more acidic $SO_4^{2-}$. Additionally, the photolysis of $pNO_3$ can also decrease the $ER$ of $NO_3^-$. As quoted from the study of Ye et al. (2023), "$pNO_3$ photolysis is surface-catalyzed and is greatly affected by the physicochemical properties of aerosol particles, such as $pNO_3$ loading, chemical composition and particle size". This may help explain the variation of the difference in $ER$ of $NO_3^-$ at A-2 and A-7.

Since this study mainly focuses on the evolution of BrC, we prefer to place the content related to $NO_x$ and $SO_2$ in the supplemental materials so that the main text can concentrate more on BrC-related topics. The discussion on $NO_x$ and $SO_2$ is mainly intended to assist in proving the occurrence of the aging process. To address the reviewer's concerns, we now add more discussion on the variation in the $NO_3^-$ $ER$: "… Similarly, $Cl^-$ depletion can also result from acid replacement by stronger acids such as $H_2SO_4$ and $HNO_3$ (Wang et al., 2019) or reactions with organic acids (Laskin et al., 2012), leading to a low $ER$ value of $Cl^-$ (i.e., $ER < 1$). **Moreover, the cycle between $NO_x$ and their oxidative reservoir ($NO_z$), has a significant impact on the $NO_2/NO_z$ ratio, and the photolysis of particulate nitrate ($pNO_3$) is proposed as a potentially important mechanism influencing the partitioning between $NO_x$ and $NO_z$ (Ye et al., 2023). However, the photolysis rate constant of $pNO_3$ is highly variable and can be greatly affected by aerosol properties, including $pNO_3$ loading and particle size (Ye et al., 2017; Andersen et al., 2023). This may also help explain the observed differences in the $ER$ of $NO_3^-$ between A-2 and A-7.**"

L173: It is not at all clear how "this could also explain the low ER value of $Cl^-$ " Detail is needed. Especially given that the literature generally supports the replacement of non-volatile (e.g. sea salt) chloride. A reference would at least help. But also some further quantitative discussion. Can displacement explain the changes?

Sorry for the confusion. The $ER$ can indicate whether a species has undergone net production ($ER > 1$) or loss ($ER < 1$). Similar to $NO_3^-$, $Cl^-$ can also be displaced by

stronger acid or depleted due to reactions with organic acids, which explains its *ER* being less than 1. However, since this study simulates the aging of biomass burning smoke, the reaction system is quite complex. It is difficult to isolate and quantify changes in individual processes as the response is non-linear. To make it clearer, we revise the text and add references: "… Similar trends of $SO_4^{2-}$ and $NO_3^-$ from A-2 to A-7 have also been reported in other studies (Guo et al., 2022), and are attributed to the replacement of $NO_3^-$ by more acidic $SO_4^{2-}$. **Similarly, $Cl^-$ depletion can also result from acid replacement by stronger acids such as $H_2SO_4$ and $HNO_3$ (Wang et al., 2019) or reactions with organic acids (Laskin et al., 2012), leading to a low *ER* value of $Cl^-$ (i.e., *ER* < 1).** …"

Fig. 1b: I very strongly encourage removing the "slope" lines that presume to indicate some chemical process. These lines have many, many assumptions baked in. Specifically, they assume that the end member meets at H:C = 2 and O:C = 0. This is true only for very specific VOCs. The inclusion of lines is misleading in terms of process. There is no evidence that these lines indicate anything about process for the samples measured. Moreover, the discussion of how aging affects things on L181 does not make sense. The change needs to be considered relative to the starting condition. In this case, the H:C increases slightly while the O:C increases with aging. This is not a move from "a slope of -2 in fresh smoke to the region with a slope of -1 upon aging." These slopes are not meaningful here. The relevant situation is that the H:C increased slightly while the O:C increased more, and thus corresponds to a "slope" of about +0.1. Interpreting this in terms of specific chemical groups cannot be done unless one knows the identity of the VOC precursors that lead to SOA formation. This discussion must be revised to focus on actual physical and chemical changes, not arbitrary lines.

Thank you for the nice suggestion. We have revised the figure to de-emphasize these lines. We also revise the text accordingly: "… The H/C and O/C values of fresh OA are 1.44 and 0.33, respectively, indicating a low carbon oxidation state ($OS_c \sim -1$). Compared to fresh OA, the O/C ratios progressively increase to 0.45 for A-2 and 0.57 for A-7, accompanied by an $OS_c$ approaching 0, which is attributed to the formation of oxygenated OA. The H/C ratios also show an increase (1.52) for A-2 but slightly decrease for A-7 compared to A-2. The evolution of H/C depends on the precursor, and in general it decreases with OH exposure due to the hydrogen loss from the C=O bond formation (Lambe et al., 2013). However, Hennigan et al. (2011) also reported a slight increase in H/C ratios. Li et al. (2023) attributed the discrepancy to the abundance of $C_2H_3O^+$ in fresh smoke, where low $C_2H_3O^+$ levels lead to an increase in m/z 43 during initial aging process."

Fig. S4: I find the "stack diagram" very difficult to interpret. It is extremely challenging to determine how the different aging conditions compare with each other from such a graph. Moreover, the % values seem wrong (too small) as they will not sum to unity.

We agree with the reviewer that the stacked plot corresponding to the left Y-axis are difficult to discern changes from fresh to aged m/z signal. So we added the ratio of the

aged to fresh signal, which is shown on the right Y-axis. A ratio greater than 1 indicates that an increase in the signal during aging, for example at m/z=28 and 44. Additionally, the use of "%" in the title was incorrect and has been removed.

L180: The ToF-ACSM does not allow for direct determination of H:C and O:C ratios. They must be inferred. The authors should state the assumption they made in translating the actual measurements to H:C and O:C.

Thanks for the suggestion. Yes, the estimation of those ratios is based on the high-resolution fragments, and their uncertainties have been discussed elsewhere (Canagaratna et al., 2015). We now revise the text to add the description and the corresponding reference: "… To further describe the implied changes in OA composition, the H/C and O/C ratios for both fresh and aged smoke are distributed on the Van Krevelen diagram in Fig. 1b. The ratios are estimated based on high-resolution fragments (i.e., m/z of 43 and 44) using calibration factors from Canagaratna et al. (2015), which derived these factors from sampling standards, assuming that the elemental composition of the original species corresponds to the averaged ion composition across the mass spectrum. …"

Section 3.2 and PMF: Some justification for the choice of 3 factors should be provided. Why 3 (versus 2 or 4 or 5)?

Thanks for the suggestion. We actually have run the PMF model with the number of factors ranging from 2 to 7. The selection of the optimal number of factors is based on

Q values as well as the inter-profile/time series correlations, etc., with some results

shown in Figure R2, and takes into account the interpretability of the factors. The Q/Qexceped ratios decrease with increasing number of factors (P). The decreasing rate of Q/Qexceped ratios is significantly lower when P >4, which implies P=4 may be the best choice. However, the correlation between factor profiles as well as their temporal correlation of P = 4 is too high compared with the condition of P=3. Therefore, we chose P=3 for final results. We now revise the text to briefly mention the reason: "To further explore the evolution of different OA components, we classified total OA from both fresh and aged smoke into different categories based on the PMF model. We ran the model with the number of factors ranging from 2 to 7 and chose three factors for the final results based on optimal fit and interpretability (Fig. 2). …"

[Figure]

Figure R2. Key diagnostic plots of the PMF results. (a) Q/Q$_{expected}$ as a function of number of factors (P); and cross-correlation coefficients (R) of the time series and spectral profiles among the PMF factors: (b) P = 3 and (c) P =4.

OOA: Further discussion of the relatively large contribution of OOA with zero aging seems needed. Certainly a factor is just a mathematical construct and not necessarily an actual thing. However, the authors discuss OOA as resulting from SOA formation. But if OOA equates to SOA formation, why does it make up to 40% of the OA with zero aging? Some consistency in the discussion would be welcome.

Thanks for the suggestion. Oxygenated OA (OOA) with relatively high O/C is generally associated with secondary formation but can also be found in fresh plumes. Especially, OA from biomass burning typically exhibits higher O/C ratios compared to other sources due to the high oxygen content of lignocellulosic materials and relatively lower combustion efficiency (Aiken et al., 2008; Heringa et al., 2011; Avery et al., 2023). This may explain that fresh OA also has some OOA. To make it clearer, we add more discussion in the text: "… This is actually a significant feature distinguishing biomass sources from other sources (Aiken et al., 2008). Biomass is naturally rich in oxygen-containing compounds, such as cellulose, hemicellulose and lignin. And those oxygen-containing structures partially break down and release oxygen-rich substances into the gas or aerosol phase during combustion. Incomplete combustion during ignition and the burnout stages can also produce oxidized OA with relatively high O/C ratios (Heringa et al., 2011; Avery et al., 2023). Previous studies have reported O/C ratios of 0.2–0.6 in fresh biomass burning smoke (Heringa et al., 2011; Fang et al., 2017; Ma et al., 2019; Li et al., 2023), significantly higher than those from traffic exhaust (around 0.02–0.19) (Chirico et al., 2010; Dallmann et al., 2014; Collier et al., 2015). …"

L218: Please clarify how a smaller particle distribution spread (sigma) means that the particles have a relatively uniform morphology or density. What does density have to do with the spread? Similarly morphology?

Sorry for the confusion. A small sigma typically indicates that the particle sizes are more cluster around the mean. We now delete the sentence to avoid overinterpretation.

L222: Often in OFR's there is substantial nucleation, given the fast oxidation conditions. The authors present maximum diameter and spread results, but do not provide any information on what the actual size distributions look like (are they, for example, log normal) or whether there is nucleation.

Thanks for the suggestion. We agree with the reviewer that the particle number size distribution after PAM aging is bimodal lognormal distribution due to large amount of nucleation, as shown in Figure R3. The revised text in line 237 now reads: "… The particle number size distribution of freshly emitted biomass smoke exhibits a unimodal lognormal distribution. After PAM aging, it exhibits a bimodal pattern, with an additional small-particle mode (around 20 nm) compared to fresh smoke, mainly resulting from nucleation. Here we only focus on the evolution of particles in accumulation mode. …"

[Figure]

Figure R3. The particle number size distribution of different biomass smoke from fresh and aged emissions.

L229: Further explanation of the statement "On the other hand, more mass is required to increase one unit particle size as the particle grows. This may also explain the observed decease in σ as particles shift to larger size along with aging (Fig. 3b)." Please clarify how the former explains the latter. Is this just noting the well-known narrowing phenomenon that occurs during growth experiments?

To make it clearer, we have revised the text to: "… This could be explained, on one hand, by the limited availability of precursors. On the other hand, as particles grow, more mass is required to increase their diameter by one unit. Consequently, smaller articles grow faster in diameter, leading to a narrower size distribution and a reduced σ upon aging (Fig. 3b)."

L235: The authors state that "Under OH exposure, the AAE decreases due to a reduction in BrC light absorption." It is not clear to me that this is a robust conclusion based on the data presented in the graphs. The bars overlap, certainly with the error bars shown. A statistical analysis seems needed to back up this statement.

Thanks for the suggestion. We conducted a Mann-Whitney U test (applicable to median difference tests), and found the difference is significant with $p < 0.05$, except for the rice samples. We now revise the text to include this information: "… Under OH exposure, the AAE decreases due to a reduction in BrC light absorption, with the *ER* of $b_{abs, BrC}$ around 0.51–0.85 and 0.63–0.90 in A-2 and A-7, respectively. **The difference is statistically significant with $p < 0.05$, except for rice samples.** The large decrease in BrC absorption at A-2 implies a significant effect of photobleaching associated with OH-driven oxidation in the PAM. …"

L238: I think that it would be great if the authors clarify what they mean by "photobleaching" as it relates to process. Do the authors really mean "degradation through heterogeneous oxidation?" Or are the authors indicating some effect of directly photolysis? I generally think of photobleaching as referring to the direct influence of

photon absorption, and I think that this is generally accepted. If the authors mean the latter (direct degradation via photons) then I cannot agree with the interpretation put forward. The photon flux and time spent in the OFR is likely insufficient to have any substantial direct impact. Instead, any changes that occur are likely a result of OH-dominated heterogeneous oxidation, coupled with changes driven by formation of SOA that has different properties than the primary OA.

Sorry for the confusion. "photobleaching" here refers to the processes by which BrC undergoes chemical transformations due to sunlight, leading to a reduction its light absorption. These include direct photolysis and photooxidation, such as reactions with OH radicals. In this study, the photobleaching is associated with OH-related oxidation. To make it clearer, we have revised the text to: "… Under OH exposure, the AAE decreases due to a reduction in BrC light absorption, with the *ER* of $b_{abs, \text{ BrC}}$ around 0.51–0.85 and 0.63–0.90 in A-2 and A-7, respectively. The difference is statistically significant with p < 0.05, except for rice samples. **The large decrease in BrC absorption at A-2 implies a significant effect of photobleaching associated with OH-driven oxidation in the PAM.** …"

L239: I am not convinced that the ER differences at A-2 and A-7 for brown carbon are actually (statistically) different from each other, and therefore the idea that there is first a photodegradation process but at later times some photo-enhancement is suspect. I believe a statistical analysis is needed.

Thanks for the suggestion. The statistical test results show that the p value ranges from 0.10 to 0.55, indicating that the difference is not significant. Accordingly, we have revised the text to: "… No significant difference is observed in the *ER* of BrC absorption between A-2 and A-7, indicating a limited photobleaching effect after 2-day aging. …"

L243: There is no physical basis for an exponential relationship between AAE and BrC fraction. What is the purpose of presenting an exponential relationship? To the extent that it can be assumed that BC and BrC equal two end members the relationship should be linear. This should be placed in a physical framework.

Thanks for the comment. Since BrC generally has a higher AAE than BC, in theory, a higher contribution of BrC to light absorption should correspond to a higher overall AAE. Here we try to provide a empirical formula. This is also done by previous studies, e.g. Sun et al. (2021). Of course, parameterization may vary significantly under different conditions, but we hope that it can still offer a useful reference for future studies.

L250: Any time that MAE is mentioned the wavelength should be included. I encourage simply using a subscript 365nm.

Sorry for the confusion. In fact, we have stated at line 156 that the light absorption and MAE in this study refer to the wavelength at 370 nm.

L255: The authors state "For OA at A-2, the reduction in MAE compared to the fresh

smoke is dominated comparably by both the changes in $b_{abs, BrC}$ and OA mass." This needs clarification. How is it known that it is dominated "comparably" by both these things? And then how do the authors know that at A7 "the reduction in MAE is mainly driven by the increasing in SOA with weak absorption." This is simply stated but has not been demonstrated.

Thanks for the comment. We now add detailed values to support the statement: "… The change in MAE is associated with the changes in both BrC absorption and OA mass concentrations. In other words, it is due to the combined effect of bleaching and the formation of SOA with weak absorption. Compared to fresh smoke, the $b_{abs, BrC}$ at A-2 is reduced by about 15%–49%, accounting for roughly half or more of the change in MAE. This suggests that the reduction in MAE at A-2 is dominated by both $b_{abs, BrC}$ and OA mass. Moreover, the MAE of rice straw, corn straw and soybean straw at A-7 is similar to that in A-2, which may imply the resistance of BrC to photobleaching after the first few days of aging. Similarly, Zhao et al. (2022) also reported that 49% to 67% of the initial MAE of rice straw remained after equivalent 9 days of aging."

L264: Please clarify how ER's are calculated for each OA factor. It is not readily apparent.

Sorry for the confusion. *ER* is calculated for the absorption of each OA factor. The calculation of *ER* of BrC absorption is similar to the equation for ER of different chemical components. To make it clear, we add the description at the end of the method section: "… The *ER* of BrC absorption was also calculated using a similar equation as for chemical components, with X representing the absorption of BrC at 370 nm and the BC term substituted by absorption at 880 nm."

L271: I appreciate the authors considering the relationship between MAE and size, but it's not clear that this makes sense when considered separately from the evolving particle composition that comes with particle growth. Moreover, the authors should likely use the mass-weighted diameter, not the number-weighted diameter, when considering any MAE-size relationship, since absorption is driven by mass. Really, it seems like what the authors want to do is an optical closure study where they use the absorption and size distribution measurements to derive the value of k. They can then consider whether k is changing. The imaginary component of the refractive index is a conserved physical property, unlike the MAE. As it stands, I do not think that the authors can confidently make the conclusions they do in this paragraph. Lastly, the linear fit line on Fig. 5d should be removed. It has no physical basis and there is no reason to believe that it will prove broadly applicable.

Thanks for the suggestion. We accordingly updated the figure using mass-weighted diameter (Figure R4). The conclusion remains essentially unchanged. We also delete the fit line, which was plotted during the initial data exploration and is not discussed in the text.

We agree with the reviewer that MAE is sensitive to the size distribution. Since particle size increases during the aging process, our original intention was to examine whether

this growth significantly affects the change in MAE. The calculations show that the observed change in MAE is mainly driven by changes in the imaginary component of the refractive index $k$. This is consistent with the discussion in the preceding section, where aging was shown to degrade strongly light-absorbing OA components while OA with weak absorption was formed, leading to a decreased $k$ for total OA.

[Figure]

Figure R4. The relationship of MAE at 370 nm with particle diameter ($D_p$), the blue, red and orange dots represent the peak sizes of the particle mass size distribution from fresh (F), 2-days (A-2) and 7-days (A-7), respectively. The MAE dependence on particle size is calculated using the Mie model for pure BrC (gray dashed lines).

Fig. 5b: Which laboratory and field observations are included here? It is not at all clear, and therefore not reproducible. Are these from Table S4 and S5? If so, how have the authors addressed the issue of wavelength differences? And which values are selected from this long list? Just those with BBOA in the name? Again, it is not clear, which makes the comparison less meaningful.

Sorry for the confusion. Sources of the data used in Fig. 5b are listed in Table S4 and S5. We now add this in the caption of the figure. To ensure wavelength consistency, we now update the figure and the related discussion only using data measured between 365 and 405 nm (and delete the others). Given this narrow range, the bias introduced by wavelength difference is assumed to be negligible. In addition, for the laboratory experiments of biomass burning, the data is for total organic aerosol in either fresh and aged smoke. For the field observations of ambient samples, we only select results for BBOA and OOA.

Fig. S5: Please report the R2 values associated with the fits. Also, please justify why it makes sense to merge the aged and non-aged samples into the same dataset when it comes to a linear fit given that the authors have argued that the composition has changed. And, what is the dashed oval? The authors favored data? Were some of the points excluded? If so, why, and how is this justified? (I don't believe that it is justified.).

Thanks for the suggestion. We have added $r^2$ in Figure S5. Additionally, we aim to construct a relationship between MAE and BC/OA that accounts for both fresh and aged states, as done in the study by (Saleh et al., 2014). Furthermore, since the MAE of one

biomass type deviates significantly (may be due to biomass types differences), we will only use the data within the dashed oval (95% confidence interval) for fitting.

L291: The idea that the derived slope of 8 is "within the range (~0.63-22)" is not especially meaningful. This is a HUGE range. Also, why are only these two studies cited? These seem randomly selected from the broad literature. The comparison to the literature here, and really throughout, is fairly weak in the sense that it often seems extremely selective. Particle growth? Nucleation? Particle size relationship with MAE?

We agree with the reviewer that the range is large, due to different biomass types, combustion efficiency, temperature and other conditions. The main purpose here is to provide more data for future study. Currently, there are very few studies reporting the relationship between MAE and BC/OA ratios. We found only these two references. To avoid confusion, we revise the text: "… 
[revised manuscript text omitted]